# TAILORED PRIMITIVE INITIALIZATION IS THE SECRET KEY TO REINFORCEMENT LEARNING

## ABSTRACT

Reinforcement learning (RL) has emerged as a powerful paradigm for enhancing the reasoning capabilities of large language models (LLMs). While RL has demonstrated substantial performance gains, it still faces key challenges, including low sampling efficiency and a strong dependence on model initialization: some models achieve rapid improvements with minimal RL steps, while others require significant training data to make progress. In this work, we investigate these challenges through the lens of reasoning token coverage and argue that initializing LLMs with diverse, high-quality reasoning primitives is essential for achieving stable and sample-efficient RL training. We propose Tailor, a finetuning pipeline that automatically discovers and curates novel reasoning primitives, thereby expanding the coverage of reasoning-state distributions before RL. Extensive experiments on mathematical and logical reasoning benchmarks demonstrate that Tailor generates more diverse and higher-quality warm-start data, resulting in higher downstream RL performance.

## 1 INTRODUCTION

Large Language Models (LLMs) have demonstrated remarkable reasoning capabilities across a broad range of application domains, including mathematical problem solving (Yu et al., 2023a; Yue et al., 2025b; Zeng et al., 2025b; Shen et al., 2025b), code generation (Xia et al., 2024; Yang et al., 2024b), and complex decision-making in agentic tasks such as API calling (Liu et al., 2024b; Prabhakar et al., 2025), autonomous driving (Li et al., 2024; Wei et al., 2024), and robotics (Yu et al., 2023b; Team et al., 2025a). Reinforcement learning (RL) has emerged as a promising paradigm for enhancing reasoning abilities by exploiting feedback from environments and leveraging verifiable reward signals (Ouyang et al., 2022; Wen et al., 2025). Such training approaches have been shown to improve model performance through the generation of long chains of thought (CoTs) (Wei et al., 2022) and test-time scaling (Yu et al., 2025c), thereby producing outputs of higher quality.

Although RL has demonstrated strong capabilities, it still faces significant challenges. First, RL suffers from low sampling efficiency (Haarnoja et al., 2018; Du et al., 2019; Shi & Chi, 2024), a limitation that is further exacerbated in the context of LLMs due to their large parameter space and the high computational cost associated with policy rollouts and updates (Sun et al., 2025; Zheng et al., 2025). Second, empirical studies (Gandhi et al., 2025) show that different LLMs respond inconsistently to RL: While some exhibit substantial performance gains, others show minimal or no improvement. This suggests that successful RL training on LLMs is highly sensitive to model initialization (Gandhi et al., 2025; Yue et al., 2025a; Cen et al., 2025; Chen et al., 2025c). To address these challenges, one line of research focuses on improving RL algorithms through dynamic

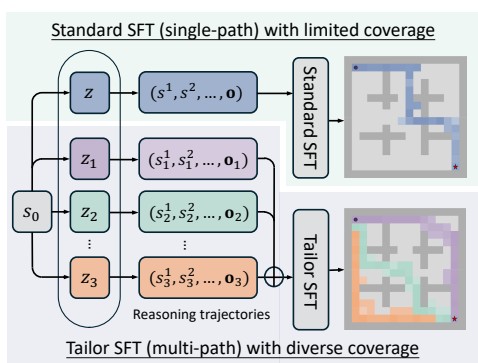

Figure 1: Coverage comparison. The goal in the maze refers to the correct answer, and the trajectories refer to the thinking tokens.

sampling (Yu et al., 2025c) and data selection techniques (Zheng et al., 2025; Sun et al., 2025). Another direction adopts a data-centric perspective, emphasizing enhancements to data quality (Yu

et al., 2025a). Prior data-centric efforts have identified emergent behavioral motifs, such as self-verification and reflection, that correlate with successful RL training (Gandhi et al., 2025). However, these patterns have typically been identified within narrowly scoped domains or tasks.

Our key insight is that successful RL training in LLMs requires initializing models with *reasoning primitives* that offer high thinking trajectory coverage. Specifically, reasoning primitive refers to reasoning patterns (*e.g.,* bottom-up reasoning, top-down reasoning, etc.), which fundamentally govern the reasoning token distribution. Most existing warm-start pipelines, where supervised fine-tuning (SFT) precedes the RL stage, include only a limited set of primitive patterns, such as rule-based traces or distilled target behaviors from the SFT dataset. However, these often exhibit low coverage of the reasoning token distribution, as illustrated in Figure 1, leading to inefficient exploration and slower RL improvement. To overcome this limitation, we propose the Tailor (**T**ask-**A**dapt**I**ve, **L**earning-**P**rimitive-**O**riented **R**einforcement) finetuning pipeline, which automatically discovers diverse reasoning primitives. Our contributions are summarized as follows:

**1. Analysis of reasoning primitive coverage.** We investigate the role of reasoning primitive diversity in warm-start RL for LLMs, shifting the focus from hand-crafting demonstrations with specific reasoning styles.

**2. The Tailor finetuning pipeline.** We introduce the Tailor pipeline, which automatically curates a reasoning-diverse SFT dataset to better initialize models for subsequent RL training.

**3. Comprehensive experimental evaluation.** We evaluate Tailor on math and logical reasoning tasks across multiple LLM models. Our experiments and ablation studies show that Tailor improves the quality and diversity of reasoning demonstrations, leading to a stronger downstream RL.

## 2 RELATED WORK

**LLM Reasoning.** Reasoning models were first conceptualized in the OpenAI series models (Jaech et al., 2024), referring to LLMs that leverage test-time scaling by generating long chains of thought (CoTs) before final answers to improve output quality (Guo et al., 2025; Team et al., 2025b). The success of OpenAI-o1 (Jaech et al., 2024) and DeepSeek-R1 (Guo et al., 2025) demonstrated that large-scale RL can incentivize reasoning capabilities in LLM training. Considerable efforts have been devoted to developing RL algorithms, such as VinePPO (Feng et al., 2023), Reinforce++ (Hu, 2025), and DAPO (Yu et al., 2025c), to advance the frontier of reasoning models. RL-based training has also been adopted in a variety of domain-specific LLMs in addition to general-purpose models (Liu et al., 2025; Shen et al., 2025a; Chu et al., 2025; Nguyen et al., 2025). Furthermore, LLM agents apply RL to enhance reasoning in textual tool use and multi-step planning (Song et al., 2025; Chen et al., 2025b; Jin et al., 2025; Qian et al., 2025), mobile and web environments (Chen et al., 2025a; Qi et al., 2024), and code generation (Wei et al., 2025; Zeng et al., 2025a; Pan et al., 2024).

**Data-Centric Methods for RL Training.** Data-centric approaches focus on improving the quality of training data rather than modifying the training algorithms (Zhou et al., 2023; Li et al., 2025; Guha et al., 2025; Yu et al., 2025b; Liang et al., 2025). Several works (Hong et al., 2023; Yao et al., 2024; Lee et al., 2024b) aim to shift the behavior policy distribution to facilitate more effective RL training (Yu et al., 2025a). Within the warm-start RL pipeline, where SFT precedes RL tuning, recent studies have shown that SFT induces coarse-grained changes in the LLM's thinking pattern distribution (Fu et al., 2025), and that RL post-training tends to amplify patterns learned during SFT (Zhao et al., 2025). These findings highlight the critical role of SFT demonstrations (Yan et al., 2025; Ma et al., 2025), which serve as a "format teacher" (He et al., 2025) to guide policy rollouts and exploration during RL. Our method falls within the data-centric category, focusing specifically on curating the SFT dataset to better prepare LLMs for downstream RL training.

**Reasoning Primitives.** Primitives are often used to represent to capture trajectory features (Goyal et al., 2020; Peng et al., 2019). In the context of LLM reasoning, the internal thinking patterns manifested in model completions have been referred to as reasoning primitives (Li et al., 2025), behaviors (Zhao et al., 2025; Cen et al., 2025), or reasoning strategies (Qu et al., 2025). Prior works have identified specific primitives, such as reflection and backtracking, as correlates of effective test-time scaling and RL performance improvements (Yeo et al., 2025; Shen et al., 2025b; Kim et al., 2025). Additionally, by comparing models that show large versus marginal gains during RL, Gandhi et al. (2025) identified four key primitives: verification, backtracking, backward chaining, and subgoaling, which are critical to RL success. In contrast to these studies, which focus on an-

alyzing specific reasoning patterns, we explore how to automatically discover novel primitives and investigate how the diversity and quality of reasoning primitives affect the effectiveness of RL.

## 3 PRELIMINARY

**Markov Decision Process (MDP).** We model the reasoning and action process of a large language model (LLM) under reinforcement learning as a Markov Decision Process (MDP) (Puterman, 2014), defined by the tuple $M = (\mathcal{S}, \mathcal{A}, P, r, s_0)$. Here, $\mathcal{S}$ denotes the state space, where each state $s \in \mathcal{S}$ encodes the current reasoning context of the LLM, including the history of reasoning tokens. The initial state $s_0 \in \mathcal{S}_0$ corresponds to a query from the query set $\mathcal{S}_0$. The action space $\mathcal{A}$ consists of all possible reasoning steps, where an action $a \in \mathcal{A}$ represents either an intermediate reasoning token or the final answer. The transition function is deterministic and defined as: $P(s_{t+1} \mid s_t, a_t) = \mathbb{I}\left[s_{t+1} = [s_t, a_t]\right]$, where each new state is formed by appending the chosen action to the current state. The reward function $r : \mathcal{S} \times \mathcal{A} \to \mathbb{R}$ specifies the immediate reward received upon taking action $a$ in state $s$. A policy $\pi_\theta : \mathcal{S} \to \mathcal{A}$ maps each state to a probability distribution over actions. A trajectory $\tau$ is defined as a sequence of states $\tau = (s_1, \ldots, s_t, \ldots, s_{T-1}, \mathbf{o})$, where $s_1$ through $s_{T-1}$ represent intermediate reasoning steps (thinking tokens) and $\mathbf{o}$ is the final answer. $T$ denotes the number of steps.

**Reinforcement Learning Fine-Tuning.** The goal of Reinforcement Learning Fine-Tuning in LLMs is to optimize the expected reward over a set of queries $\mathcal{S}_0$:

$$\max_\theta \mathcal{J} = \mathbb{E}_{s_0 \sim \mathcal{S}_0, \, s \sim \pi_\theta} \left[ \sum_{t=1}^{T} r(s_0, \tau^{(\leq t)}) \right], \quad \text{where } r(s_0, \tau) = \mathbf{1}(\mathbf{o} = \mathbf{o}_{\text{gold}}), \tag{1}$$

where $\mathbf{o}_{\text{gold}}$ is the ground-truth answer to the query. In this work, we primarily adopt a rule-based outcome reward that assigns a binary signal to the final answer tokens in the generated output. We study the objective function of *KL-regularized clipped policy optimization*, which incorporates regularization toward a reference policy, applies clipping to the policy ratio, and optimizes the following surrogate objective:

$$\mathcal{J}_{\text{RL}}(\mathcal{S}_0; \theta) = \mathbb{E}_{s_0 \sim \mathcal{S}_0, \tau \sim \pi_\theta(\cdot|s_0)}$$

$$\frac{1}{N} \sum_{i=1}^{N} \left[ \min \left\{ \frac{\pi_\theta(\cdot|\tau^{(<t)})}{\pi_{\text{old}}(\cdot|\tau^{(<t)})} A_i, \text{clip}\left( \frac{\pi_\theta(\cdot|\tau^{(<t)})}{\pi_{\text{old}}(\cdot|\tau^{(<t)})}, 1 - \epsilon_1, 1 + \epsilon_2 \right) A_i \right\} - \beta D_{\text{KL}}(\pi_\theta \| \pi_{\text{ref}}) \right], \tag{2}$$

where $\epsilon_1, \epsilon_2$ are the clipping ratios, $\beta$ is the KL regularization coefficient, and $A_i$ is the advantage term determined by the specific RL algorithm. This objective is used in many RL frameworks such as GRPO (Shao et al., 2024) and DAPO (Yu et al., 2025c).

**Warm-Start RL.** This work focuses on the warm-start RL pipeline, which first applies Supervised Fine-Tuning (SFT) followed by RL training (Shao et al., 2024). The SFT stage enables LLMs to follow formatting instructions, become familiar with the dataset, and acquire initial reasoning capabilities within the target domain using a demonstration dataset $\mathcal{D}_{\text{SFT}}$. During SFT, the policy $\pi_\theta$ is trained by minimizing the negative log-likelihood loss:

$$\min_\theta \mathcal{L}_{\text{SFT}}(\mathcal{D}; \theta) = -\mathbb{E}_{(s_0, \tau, a) \sim \mathcal{D}_{\text{SFT}}} \left[ \sum_{t=1}^{T} \log \pi_\theta(a_t \mid s_0, \tau^{(<t)}) \right] \tag{3}$$

For simplicity, we use the term SFT model to refer to the model after SFT.

## 4 METHOD

### 4.1 REASONING PRIMITIVE

In warm-start RL, LLMs initially adopt the thinking token distribution from the SFT dataset and progressively refine their reasoning patterns in RL through interaction with verifiers. Empirical studies (Gandhi et al., 2025; Cen et al., 2025) show that the distribution of thinking tokens in the SFT dataset strongly influences downstream reasoning performance: when warm-starting from two SFT datasets with different reasoning chain patterns, the resulting RL performance can diverge dramatically, even though both patterns are valid and interpretable to humans. To investigate this

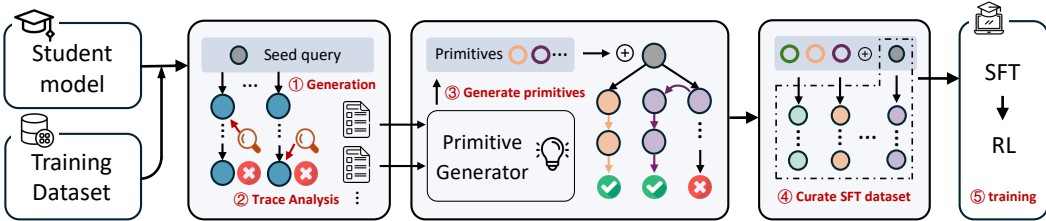

Figure 2: Overview of the Tailor finetuning pipeline.

phenomenon and identify better CoT patterns, we introduce the notion of *primitives*, which control the trajectory-wise distribution $\rho$ of thinking tokens to model textual reasoning patterns. Formally, we augment the MDP tuple $M' = (\mathcal{S}, \mathcal{A}, P, r, s_0, z)$ with a reasoning primitive $z$ (Qu et al., 2025):

$$\rho(\pi_\theta \mid z) = \mathbb{E}_{s_0 \sim \mathcal{S}_0} \prod_t \pi_\theta(a_t \mid s_t, z) \, \mathbb{I}[s_{t+1} = [s_t, a_t]], \tag{4}$$

The distribution of thinking tokens $\rho$ is influenced by the primitives $z$. In the context of LLMs, primitives $z$ can correspond to prompt instructions that are combined with the query $s_0$ to guide subsequent reasoning. We provide several textual examples of reasoning primitives in Figure 3. For instance, when constructing reasoning chains for a math problem, different primitives such as *Top-Down* and *Bottom-Up* reasoning, or strategies for self-verification and error recovery, can be applied. Primitives also encompass broader behaviors such as *reflection* and *backtracking*, which enable the model to detect and recover from failures during the reasoning process. This concept naturally extends beyond mathematics to other reasoning domains, such as software engineering tasks (Zhang et al., 2025), where primitives emerge from variations in agentic and prompt designs.

Primitive initialization plays a key role in warm-start RL: due to the regularization term and policy clip mechanism in the RL objective (2), LLMs are hard to automatically discover novel primitives themselves during the RL process (Zhao et al., 2025), especially when we are training specialized LLMs with limited capability and pre-training data distribution. Prior works have identified specific patterns, including *backtracking*, and *reflection* (Gandhi et al., 2025), that have a correlation with performance improvement in the subsequent RL stage. We interpret the success of specific primitives as the increase in the thinking token distribution coverage, as *reflection* and *backtracking*, which means revisit of previous context, implicitly increasing the probability to explore other thinking states.

### 4.2 TAILOR PIPELINE

Building on the above analysis, we argue that initializing LLMs with broad thinking token coverage leads to stronger exploration capabilities and higher RL performance. Our objective is therefore to discover a more diverse set of reasoning primitives, thereby expanding the coverage of thinking tokens and improving both the potential and sampling efficiency of the subsequent RL process. Beyond diversity, the quality of primitives is equally critical to the efficiency of RL. As noted by (Qu et al., 2025), high-quality primitives enable LLMs to achieve superior performance on specific problem sets. We assess the

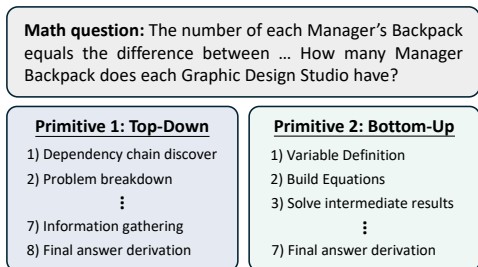

Figure 3: Examples of reasoning primitives.

quality of primitives from two perspectives: (1) they should be consistent with and aligned to the target domain tasks, and (2) they should be easy for LLMs to learn, meaning that the SFT dataset distribution remains close to the pre-training distribution.

Having established the importance of diverse and high-quality reasoning primitives, we now introduce our method for initializing LLMs prior to RL. To meet the objectives of diversity and quality, we propose the **T**ask-**A**dapt**I**ve, **L**earning-**P**rimitive-**O**riented **R**einforcement (**Tailor**) fine-tuning pipeline, illustrated in Figure 2. The key components of our approach are described as follows:

**Demonstration Trace Analysis.** The first step is to prompt the instructed student LLMs to generate completions on a small training data subset and label the answers using verifiers. These demonstrations reflect thinking tokens close to the student models' pre-training distribution and exhibit a wide range of failures and errors. We then employ an LLM-based teacher model[1] to analyze these traces and propose corrections and alternative reasoning paths toward the correct answers when failures are detected. This step is designed to uncover diverse repair-oriented reasoning patterns that remain close to the student models' pre-training distributions, making them easier to adopt and follow.

**Reasoning Primitive Synthesis.** Building on the summarized traces from the previous step, we synthesize reasoning primitives by prompting the teacher model to generate corresponding primitives $z$ that guide LLMs to produce failure-recovery reasoning patterns during generation. Given that the observed failures span a wide range of types, we are able to curate a broad collection of reasoning instructions to support the subsequent SFT dataset curation.

**Tailor SFT Dataset Curation.** After obtaining the set of reasoning primitives $Z$, we prompt the LLM teacher model to generate reasoning traces that explicitly follow each primitive.

After curating the Tailor SFT dataset, we fine-tune the student models on it and subsequently apply RL. A simplified overview of the Tailor process is provided in Algorithm 1, and additional details are included in the Appendix A.

---

**Algorithm 1** Tailor Finetuning Pipeline

**Input:** SFT seed dataset $\mathcal{D}_s$, subsets of RL dataset $\mathcal{D}_{RL}$, student model $\pi_\theta$, teacher model $\pi_t$.
**Output:** Tailor RL model $\pi_{\theta'}$.

1: # Demonstration trace analysis.
2: Generate completions: $\tau \sim \pi_\theta(\cdot \mid s_0), s_0 \sim \mathcal{D}_{RL}$
3: Summarize Failures $\mathcal{F} = \{f_k\}, f_k \leftarrow \pi_t(\tau_k)$.
4: # Reasoning primitive synthesis.
5: Generate primitives $\mathcal{Z} = \{z_m\}: z_m \leftarrow \pi_t(f_m)$
6: # Tailor SFT dataset curation.
7: Initialize SFT $\mathcal{D}_{SFT} \leftarrow \emptyset$
8: **for** $(s_0, a) \in \mathcal{D}_s$ **do**
9:     Sample a primitive $z \sim \mathcal{Z}$;
10:    Curate trace: $\tau \leftarrow \pi_t(\cdot \mid s_0, z)$;
11:    Update $\mathcal{D}_{SFT} \leftarrow \mathcal{D}_{SFT} \cup (s_0, \tau)$
12: **end for**
13: # SFT & RL.
14: Perform SFT (3) on $\mathcal{D}_{SFT}$ and RL (2) on $\mathcal{D}_{RL}$.
15: **Return:** RL policy $\pi_{\theta'}$

---

**Intuition.** In the Tailor pipeline, student models often exhibit a variety of reasoning failures in their demonstration traces. These failures include skipping intermediate steps, making unsupported assumptions, or mishandling mathematical relationships. By analyzing these traces, the teacher model constructs a targeted set of repair primitives $\mathcal{Z}$ that directly address the observed error patterns. This set is not only tailored to the student model's generation behaviors and the target domain but is also enriched with diverse instructions that correspond to distinct types of reasoning errors. As a result, the dataset generated using $\mathcal{Z}$ includes a broader range of reasoning primitives with greater coverage. This increased coverage improves alignment with the task distribution and provides a more effective foundation for subsequent reinforcement learning, leading to higher sampling efficiency and performance gains in warm-start RL.

## 5 EXPERIMENTS

We now evaluate the effectiveness of Tailor in improving RL performance. We begin by presenting the main results, followed by a comparison of primitive diversity, and conclude with ablation studies.

### 5.1 EXPERIMENT SETUP

**Dataset and Benchmarks.** We conduct experiments on two reasoning benchmarks: (1) iGSM (Ye et al., 2024), a grade-school math benchmark that tests mathematical and commonsense reasoning skills, containing difficulty of medium and hard tasks; and (2) KK (Knights & Knaves) (Xie et al., 2024), a logical reasoning benchmark based on dynamically generated knights and knaves puzzles. We choose these two benchmarks to assess problem-solving and reasoning capabilities in LLMs while minimizing the influence of factual knowledge retrieval and reducing the risk of data contamination from the pre-training stage (Wu et al., 2025), as the synthetic nature of iGSM and KK helps

---

[1] We deploy DeepSeek-V3-0324 as the teacher model in our experiments.

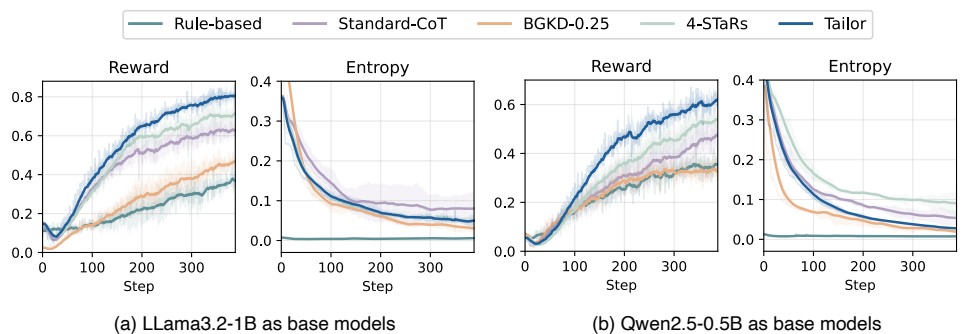

Figure 4: Training Curves of the KK tasks. We average curves with 3 random seeds.

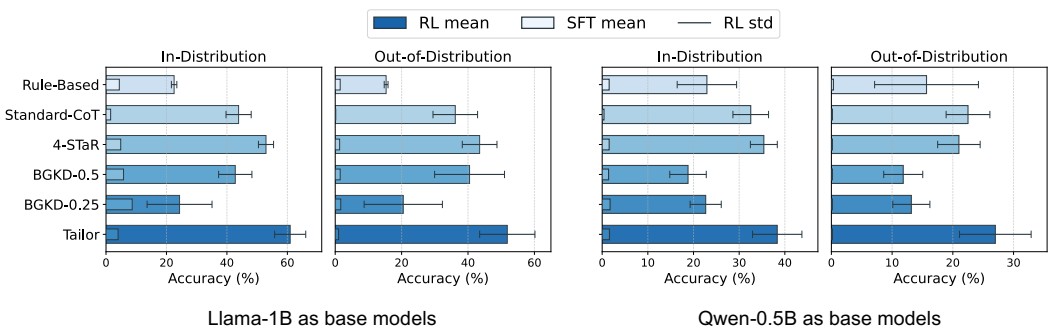

Figure 5: The evaluation results (%) on the KK reasoning tasks. We train SFT models for 4 epochs and finetune them with RL for 5 epochs. The mean value and standard deviation are calculated over 3 random seeds.

mitigate such issues (Ye et al., 2024; 2025). For both datasets, we evaluate methods in in-distribution (In-Dist) data and out-of-distribution (OOD) data. See Appendix A for more details.

**Baselines.** We compare Tailor with three types of baselines: (1) Rule-based Demonstration. Both the iGSM and KK benchmarks provide ground-truth functions to generate rule-based reasoning traces. We construct CoTs using these rule-based traces, perform SFT on the data, and then apply RL. We refer to this baseline as **Rule-based**. (2) Human-Crafted Primitives. Gandhi et al. (2025) identifies four critical STaR behaviors for RL and injects these reasoning patterns by prompting a teacher model with specialized instructions. We use their prompts to collect demonstrations and perform distillation, referring to this baseline as **4-STaR**. Additionally, we include a **Standard-CoT** baseline in this category, where teacher demonstrations are generated using CoT prompting (Wei et al., 2022). (3) Re-distillation. Chen et al. (2025c) propose a re-distillation strategy in which a trained model is used to regenerate the SFT dataset, producing data that better aligns with the model's pre-training distribution. This idea is consistent with Generalized Knowledge Distillation (GKD) (Agarwal et al., 2024), which aims to reduce the distributional gap between the finetuning dataset and the student model's pre-training distribution. We apply re-distillation to the 4-STaR SFT model to curate a new dataset, which we then combine with the original 4-STaR dataset. After SFT on this mixed data, we apply RL. We refer to this baseline as Batch-GKD (**BGKD**-$\lambda$), where $\lambda \in [0, 1]$ denotes the proportion of re-distilled data in the SFT dataset.

**Other Experiment Settings.** We evaluate our method using the Llama (Grattafiori et al., 2024) and Qwen (Yang et al., 2024a) model families. For Llama, we use Llama3.2-1B and Llama3.2-3B as base models; for Qwen, we use Qwen2.5-0.5B and Qwen2.5-3B. The demonstration dataset size for the SFT stage is set to 8,000, and the RL dataset contains 10,000 examples. Unless otherwise specified, we use DeepSeek-V3 (Liu et al., 2024a) as the teacher model with a decoding temperature of 0.5. The evaluation temperature for both SFT and RL models is set to 1.0. For RL, we adopt the DAPO algorithm (Yu et al., 2025c) with a rule-based outcome reward based on final answer

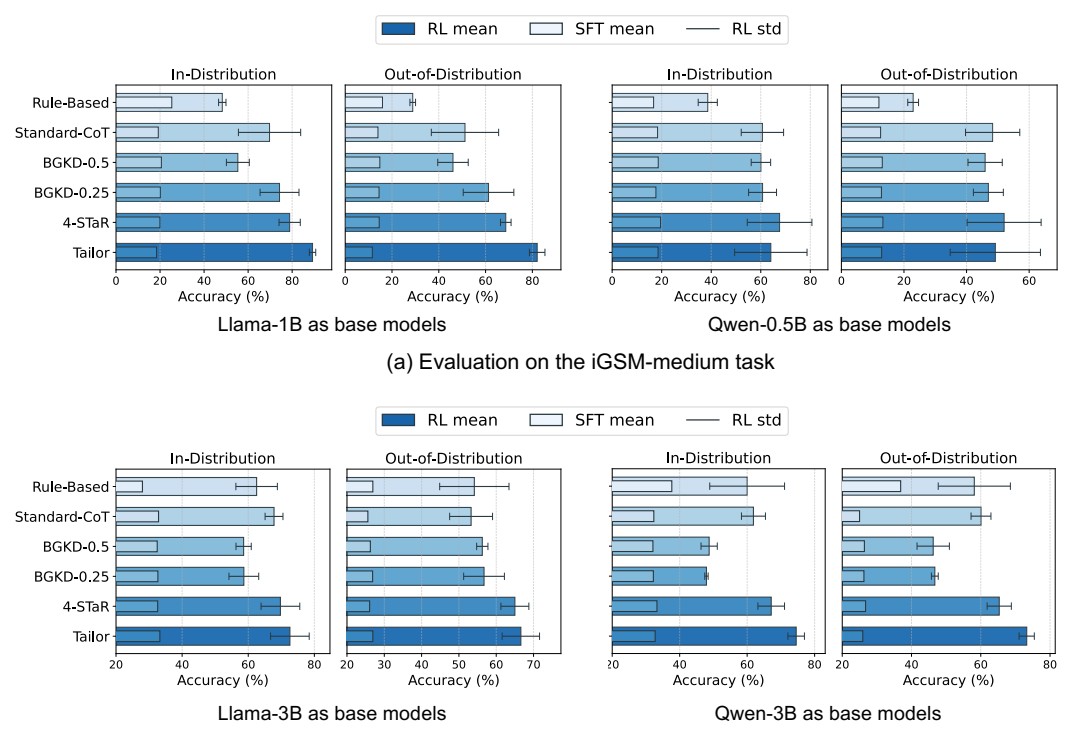

Figure 6: The evaluation results (%) on the iGSM reasoning tasks. We train SFT models for 4 epochs and finetune them with RL for 5 epochs for the iGSM-medium tasks, and finetune with RL for 100 steps for the iGSM-hard tasks. The mean value and standard deviation are calculated over 3 random seeds.

accuracy. All experiments are conducted using the Verl framework (Sheng et al., 2025). Additional details are provided in Appendix A.

## 5.2 MAIN RESULTS

We present the RL performance on the KK and iGSM tasks in Figure 5 and Figure 6, respectively. For the KK task, we simulate a practical setting in which the teacher model performs well, while the student model fails to achieve satisfactory performance. The teacher model achieves approximately 95% accuracy on the SFT dataset. We do not perform rejection sampling in this setting, as incorrect answers are rare and still provide useful learning signals (Xie et al., 2024). In contrast, for the iGSM tasks, we simulate a scenario where the teacher model performs poorly, achieving only around 25% accuracy on the SFT dataset. In this case, we apply rejection sampling during SFT data collection to filter out incorrect answers, while keeping the overall dataset size unchanged. The SFT dataset size is fixed at 8,000 for both KK and iGSM tasks.

**KK Experiments.** From Figure 5, we observe that both Llama and Qwen models struggle to achieve competitive performance after RL with rule-based CoTs. This suggests that accuracy and reasoning consistency alone are insufficient indicators of whether an SFT dataset provides good demonstrations for efficient RL. For example, rule-based annotations yield perfectly accurate and coherent reasoning traces, yet still fail to support effective RL training. These findings highlight the importance of curating SFT datasets that better prepare LLMs for RL.

Re-distillation methods (BGKD-λ) show modest performance improvements during the SFT stage. By using self-generated data, these methods reduce the distributional gap between post-training and pre-training data, resulting in more learnable patterns and improved SFT efficiency. However, BGKD still underperforms after the RL stage, as relying solely on re-distilled data diminishes the exploration benefits provided by the teacher model. In contrast, baselines such as Standard-CoT and

4-STaR achieve substantial RL performance gains by incorporating specific reasoning behaviors from teacher demonstrations. Finally, our method Tailor achieves the best overall RL performance and the largest improvements on both in-distribution and OOD evaluation sets, benefiting from the more diverse reasoning primitives it provides. As shown in Figure 4, Tailor also exhibits faster learning and higher sampling efficiency, attributed to its higher initial thinking token coverage. The entropy of the baseline methods, Standard-CoT and 4-STaRs, also remains high, but their training rewards are lower. This suggests that they may be exploring only superficial variations, such as changing individual words, rather than engaging in meaningful strategy-level exploration. As a result, the benefit for RL is limited.

**iGSM Experiments.** In the iGSM experiments, we observe that although rule-based methods enable good SFT performance, achieving top accuracy among some methods, they fail to provide a suitable starting point for efficient RL. This observation is consistent with the findings in the KK experiments and further reinforces that initial SFT model accuracy is not the sole factor influencing RL effectiveness; the underlying thinking token distribution also plays a crucial role. BGKD-$\lambda$ yields only modest improvements in RL performance. In contrast, the 4-STaR baseline enhances RL outcomes by incorporating behaviors such as *subgoaling* and *reflection*, which have been shown to correlate with successful RL. Our method, Tailor, achieves top post-RL performance by leveraging the diverse reasoning primitives introduced during the SFT stage. In experiments using Qwen-0.5B as the base model, 4-STaR performs slightly better, likely due to the limited capacity of smaller models to learn and

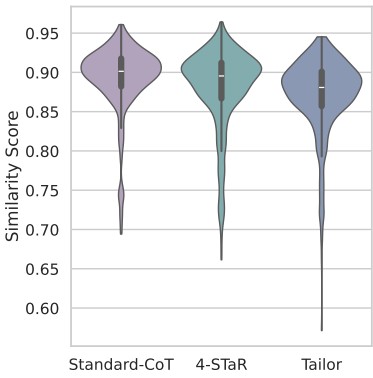

Figure 7: Primitive similarity: lower score means higher diversity.

generalize from the diverse reasoning primitives introduced by Tailor during SFT. Nevertheless, Tailor consistently demonstrates substantial RL performance gains over all other baselines across other iGSM testing datasets.

**Diversity Analysis.** To evaluate the diversity of reasoning primitives in the curated SFT datasets, we use NV-Embed-v2 (Lee et al., 2024a) to compute embeddings of the thinking tokens and calculate the average pairwise cosine similarity for responses to the same seed query. Results on the KK dataset are shown in Figure 7. We observe that 4-STaR improves diversity over Standard-CoT, as reflected by a lower median similarity and a longer tail extending toward lower values. This improvement can be attributed to the inclusion of exploration behaviors such as *reflection* and *backtracking*, which increase variation in the generated reasoning traces. Moreover, our proposed method Tailor further reduces the similarity scores compared to 4-STaR, with an even lower median and a heavier tail in the low-similarity region. This indicates that Tailor significantly enhances high-level reasoning diversity in the SFT dataset, better preparing LLMs for effective exploration during RL.

> **Takeaways**
>
> **(1)** Reasoning correctness in $\mathcal{D}_{\text{SFT}}$ alone does not guarantee RL success: although rule-based CoT achieves the highest accuracy in $\mathcal{D}_{\text{SFT}}$, it does not result in an efficient RL process.
> **(2)** Thinking-token coverage is crucial for RL initialization: Tailor increases reasoning-trajectory diversity in $\mathcal{D}_{\text{SFT}}$, enhancing exploration and improving RL efficiency.

## 5.3 ABLATION STUDY

**Ablation on Reasoning Primitives.** In our main experiments, we use 25 reasoning primitives for SFT data collection. To investigate the effect of diversity and coverage in reasoning primitives, we vary the size of the primitive set from 4 to 25 while keeping the overall SFT dataset size fixed. We perform ablations on the iGSM-medium task using Llama-1B as the base model. The results are shown in Figure 8 (a). We observe that when the primitive set is small and less diverse, the resulting RL performance is lower. This is likely because the limited primitive set covers only a small subset of effective strategies, making it difficult to generalize across a wide range of questions and hard for exploration. As the number of reasoning primitives increases to 25, post-RL performance improves, highlighting the importance of primitive coverage when preparing LLMs for RL.

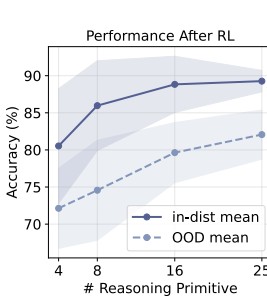 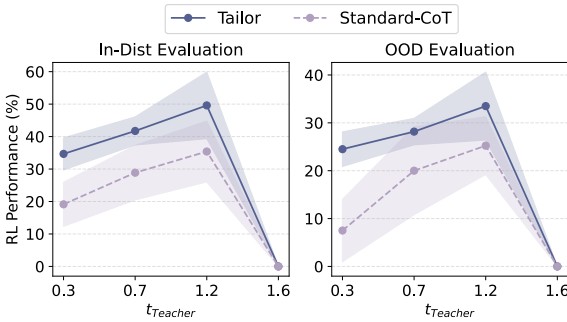

(a) Ablation on Reasoning Primitive     (b) Ablation on Teacher Model Decoding Temperature

Figure 8: Ablation Study. (a) Effect of the number of reasoning primitives in the SFT dataset. Experiments are conducted on the iGSM-medium dataset using Llama3.2-1B as the base model. (b) Effect of the teacher model's decoding temperature. Experiments are conducted on the KK dataset using Qwen2.5-0.5B as the base model.

**Ablation on Teacher Model Temperature.** Adjusting the teacher model's sampling temperature $t_{\text{Teacher}}$ is a common technique for controlling the diversity and coverage of demonstration datasets (Mukherjee et al., 2023; Hong et al., 2024). We vary the decoding temperature of the teacher model and conduct ablations on the KK task using Llama-1B as the base model. The results are shown in Figure 8 (b). We observe that as the decoding temperature increases, RL performance generally improves, suggesting that higher diversity in teacher outputs benefits downstream training. However, our method Tailor consistently outperforms the baselines across all tempera-

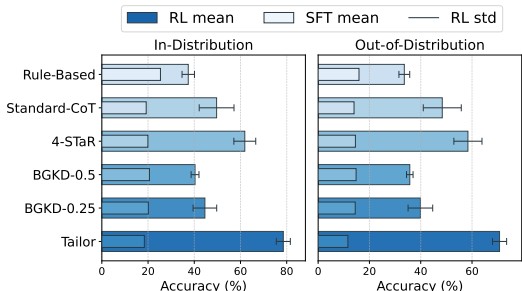

Figure 9: GRPO experiments on the iGSM-medium task with LLama-1B as base models.

ture settings, demonstrating a better balance between quality and diversity. Notably, when the temperature is set too high (e.g., $t = 1.6$), the quality of demonstrations deteriorates significantly, and the student model fails to learn reasonable traces in SFT, resulting in near-zero accuracy after RL.

**Ablation on Alternative RL Algorithms.** In addition to DAPO, we also combine our Tailor pipeline with GRPO (Shao et al., 2024) on the iGSM-medium task using Llama-1B as the base model. The results are shown in Figure 9, where we observe that the conclusions remain consistent with those from the DAPO: Tailor achieves the highest RL improvement and final performance. These results demonstrate the compatibility of our approach with a broader range of RL algorithms.

## 6 CONCLUSION

In this paper, we interpret the variation in RL performance across different initial models and the issue of low sampling efficiency through the lens of reasoning primitive quality and coverage. To address these challenges, we propose Tailor, a pipeline that discovers diverse and high-quality reasoning primitives to construct demonstration data for warming up LLMs in RL. By increasing the coverage of the thinking token distribution, Tailor facilitates faster exploration and unlocks greater performance potential during RL tuning. Extensive experiments across multiple benchmarks demonstrate that our method effectively curates a more diverse training corpus and significantly improves downstream RL performance. One limitation of Tailor is current evaluation focuses on logical and mathematical reasoning tasks. As future work, we plan to extend the pipeline to other domains such as code generation and agent-based decision-making. A potential negative impact is that misuse of our method could lead to the generation of unsafe or toxic primitives. Nevertheless, we believe Tailor sheds light on data-centric RL, emphasizing that a successful RL process depends on well-initialized models with diverse and high-quality thinking trajectory distribution.

## REPRODUCIBILITY STATEMENT

Descriptions of our method are provided in Section 4, with implementation details and prompts included in Appendices A and B. Experiments shown in Section 5 were run with multiple random seeds, and we report mean values and standard deviations across runs. We use publicly available datasets and benchmarks, with descriptions also provided in the appendix.

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

# A MORE DETAILS OF EXPERIMENTS

## A.1 EVALUATION BENCHMARK

In this section, we introduce the KK and iGSM benchmarks used for evaluation.

**KK** (Xie et al., 2024) is a logical reasoning benchmark composed of dynamically generated Knights and Knaves (KK) puzzles. In each puzzle, every character is either a Knight (who always tells the truth) or a Knave (who always lies). The puzzle presents a set of natural language statements made by the characters, referring to themselves or others. The task is to infer the identity (Knight or Knave) of each character based on these statements. Solving the puzzle requires ensuring global logical consistency: the assigned identities must be compatible with the logical implications of all statements. The input is the full puzzle text (i.e., all character statements), and the expected output is a complete assignment of identities for all characters. A prediction is considered correct only if the identities of all characters are correctly labeled. We provide an example of a query below.

---

**Query**:
A very special island is inhabited only by knights and knaves. Knights always tell the truth, and knaves always lie. You meet 7 inhabitants: Abigail, Aria, Jacob, Liam, James, Amelia, and David. In Abigail's words: "David is a knight if and only if Liam is a knight". Aria noted, "If Abigail is a knight then Liam is a knight". "Abigail is a knave" - Jacob. As Liam put it, "If Amelia is a knight then Abigail is a knight". "Jacob is a knight or Amelia is a knave," James claimed. "David is not a knight" - Amelia. David said that Jacob is a knave and David is a knight. So who is a knight and who is a knave?

---

**Rule-based CoT**:
<think>
Assume Abigail is a knight. No contradiction is found in their claim that David is a knight if and only if Liam is a knight. Assume David is a knight. No contradiction is found in their claim that Jacob is a knave and David is a knight. Jacob cannot be a knight, because this would contradict the claim of their own that Abigail is a knave. Assume Jacob is a knave. No contradiction is found in their false claim that Abigail is a knave. Assume Liam is a knight. No contradiction is found in their claim that If Amelia is a knight then Abigail is a knight. Amelia cannot be a knight, because this would contradict the claim of their own that David is not a knight. Assume Amelia is a knave. No contradiction is found in their false claim that David is not a knight. Assume Aria is a knight. No contradiction is found in their claim that If Abigail is a knight then Liam is a knight. Assume James is a knight. No contradiction is found in their claim that Jacob is a knight or Amelia is a knave. < /think>
<answer>
Thus, the final answer is boxed{Abigail is a knight. David is a knight. Jacob is a knave. Liam is a knight. Amelia is a knave. Aria is a knight. James is a knight.}
< /answer>

---

**iGSM** (Ye et al., 2024) is an infinite grade-school math problem benchmark designed to evaluate LLMs' mathematical and commonsense reasoning abilities. We apply slight modifications to iGSM to improve fine-tuning stability, including the removal of modulo operations (Cen et al., 2025). This is motivated by prior findings that base models struggle to compute modulo with high accuracy, and such capability cannot be substantially improved through training on a small SFT dataset (Ye et al., 2024). The iGSM benchmark is constructed under the assumption that ground-truth reasoning chains follow a Directed Acyclic Graph (DAG) structure. It is important to note that our proposed Tailor does not leverage any prior knowledge of this DAG representation. Implementation details, including definitions of nodes and edges, their generation process, and the natural language translation procedure, are thoroughly introduced in the original paper (Ye et al., 2024). We provide a sample query and rule-based CoT reasoning as an example below.

---

**Query**:
The number of each Grab & Go Store's Canned Peaches equals the sum of each QuickMart's Corned Beef, each Colonial Quarter's Product and each Colonial Quarter's QuickMart.
The number of each QuickMart's Canned Peaches equals 0.

---

The number of each QuickMart's Corned Beef equals 4 more than each QuickMart's Canned Peaches.
The number of each Colonial Quarter's QuickMart equals 6 more than the difference of each QuickMart's Corned Beef and each QuickMart's Product.
The number of each Vintage Architecture District's Grab & Go Store equals the sum of each QuickMart's Canned Peaches and each Colonial Quarter's Product.
The number of each Grab & Go Store's Corned Beef equals 0 times each QuickMart's Corned Beef.
The number of each Vintage Architecture District's QuickMart equals the sum of each Colonial Quarter's QuickMart and each Colonial Quarter's Product.
How many Product does each Vintage Architecture District have?

---

**Rule-based CoT**:

<think>
Let's compute the answer step by step. - According to the information given, the number of each QuickMart's Canned Peaches is 0. Let's call it D. So D = 0.
- Now, we can find the number of each QuickMart's Corned Beef. Let's call it b. Then b = 4 + D = 4 + 0 = 4.
- We can then calculate the number of each QuickMart's Product. Let's call it Y. Then Y = b + D = 4 + 0 = 4.
- We can then calculate the number of each Colonial Quarter's QuickMart. Let's call it d. Then d = 6 + (b - Y) = 6 + (4 - 4) = 6.
- Then, let's denote the number of each Colonial Quarter's Product as V. Then V = d * Y = 6 * 4 = 24.
- Then, let's denote the number of each Vintage Architecture District's Grab & Go Store as v. Then v = D + V = 0 + 24 = 24.
- Now, we can find the number of each Grab & Go Store's Canned Peaches. Let it be c. Then c = b + V + d = 4 + 24 + 6 = 34.
- We can then calculate the number of each Grab & Go Store's Corned Beef. Let it be l. Then l = 0 * b = 0 * 4 = 0.
- Now, we can find the number of each Grab & Go Store's Product. Let's denote it as r. Then r = c + l = 34 + 0 = 34.
- We can then calculate the number of each Vintage Architecture District's QuickMart. Let's call it i. Then i = d + V = 6 + 24 = 30.
- We can then calculate the number of each Vintage Architecture District's Product. Let's call it x. Then x = v * r + i * Y = 24 * 34 + 30 * 4 = 936.
Thus, the answer is 936.
</think>
<answer>
The final answer is $\boxed{936}$.
</answer>

**Difficulty Control.** Both KK and iGSM offer controllable difficulty. In iGSM, difficulty is determined by the *number of operations* required to reach the correct answer in the ground-truth rule-based reasoning trace. In KK, difficulty is controlled by the *total number of characters* (knights and knaves) involved in the puzzle. For KK, the SFT dataset includes puzzles with 4–8 characters, and the RL dataset spans 7–11. The in-distribution and out-of-distribution (OOD) evaluations use puzzles with 7–11 and 12–13 characters, respectively. For the iGSM-medium task, the SFT dataset contains problems with 15–20 operations, and the RL dataset contains 15–20. The in-distribution and OOD evaluation sets use 15–20 and 21–25 operations, respectively. For the iGSM-hard task, the SFT dataset again includes 15–20 operations, while the RL dataset contains 25–30. The in-distribution and OOD evaluation sets use 25–30 and 31–35 operations, respectively.

## A.2 TRAINING DETAILS

**SFT Training.** We perform supervised fine-tuning (SFT) on datasets curated using both Tailor and baseline methods. The key hyperparameters for SFT in the iGSM and KK tasks are summarized in Table 1. For dataset construction, we use 1,000 seed queries from the iGSM benchmark and 500 from the KK dataset.

Table 1: Configurations of SFT training

| Configurations | Value |
|---|---|
| training epochs | 4 |
| global batch size | 32 |
| learning rate | $5 \times 10^{-6}$ |
| learning rate scheduler | constant |
| padding | not removed |

**RL training.** We adopt DAPO (Yu et al., 2025c) for the main experiments using the Verl training framework (Sheng et al., 2025). The key hyperparameters for DAPO are listed in Table 2. For the GRPO experiments shown in Figure 9, we also set the overlong buffer length to 3,072 tokens and the overlong penalty factor to 1.0 to mitigate CUDA out-of-memory issues during RL training.

Table 2: Key hyperparameters for RL (DAPO) training

| Configurations | Value |
|---|---|
| training epochs | 5 |
| batch size | $128 \times 16 = 2048$ |
| sequence parallel size | 2 (actor), 1 (ref) |
| gradient clipping | 1.0 |
| entropy coefficient | 0.0 |
| learning rate | $1 \times 10^{-6}$ |
| weight decay | 0.1 |
| warmup steps | 10 |
| optimizer | Adam |
| responses per prompt | 16 |
| max response length | 4000 tokens |
| temperature | 1.0 |
| top-p | 1.0 |
| top-k | $-1$ |
| KL loss coefficient | 0.0 |
| clip ratio (low) | 0.2 |
| clip ratio (high) | 0.28 |
| remove padding | True |
| overlong penalty factor | 1.0 |
| overlong buffer length | 3072 tokens |
| rollout backend | vllm (Kwon et al., 2023) |

**RL reward verifiers.** As described in Section 3, we use an outcome-based reward in the RL process. In both the KK and iGSM tasks, the final answer is extracted via string matching. For the KK experiments, the SFT demonstration template is as follows:

```
<think>
... Thinking process ...
< /think>
<answer>
Thus, the final answer is boxed{ {name} is a {role}, ...}.
< /answer>
```

Here, name refers to the character's full name, and role refers to either Knave or Knight. To verify the correctness of the final answer, we examine each name–role pair using string matching. A completion is rewarded with $r = 1$ only if all roles are correctly assigned. Otherwise, the trace is labeled with $r = 0$.

For the iGSM task, we also use string matching in the reward function. The SFT demonstration template is as follows:

<think>
... Thinking process ...
< /think>
<answer>
The final answer is $\boxed{\{answer\}}$ .
< /answer>

where the answer is always an integer. We assign $r = 1$ if the answer matches the ground-truth value, and 0 otherwise.

**SFT curation pipeline details.** In the first step of our Tailor pipeline, where student models generate demonstrations and the teacher model analyzes the traces, we provide the teacher with three randomly selected incorrect traces and one randomly selected correct trace. The teacher is instructed to analyze the failure cases and identify the reasoning behind the correct trace. We prompt the teacher model to generate Chain-of-Thought (CoT) reasoning enclosed between special tokens, followed by the generation of reasoning primitives (i.e., prompts used to curate the SFT dataset in the next step). The prompt used in this process is provided in Appendix B.1. Additionally, we include examples of primitives in Appendix B.2.

## B PROMPTS

### B.1 PROMPTS FOR TRACE ANALYSIS AND PRIMITIVE SYNTHESIS.

> You are a large language model. Follow the instructions below carefully. Your goal is to generate instruction prompts that guide student models to produce high-quality reasoning.
> (1) Below is a correct demonstration generated by a student model for the given query. Analyze the reasoning process and identify which behaviors or strategies are particularly effective and beneficial for the model's reasoning.
> Query (Correct Case): {correct query}
> Correct CoT: {correct response}
> (2) Below are three failure cases, each consisting of the original query, the incorrect response from the student model, and the expected ground-truth answer. Analyze why each response is incorrect by referencing specific steps or reasoning patterns that led to the failure. Based on this analysis, provide instructions on how to identify and revise the failed demonstration to arrive at the correct answer.
> Failure Case 1:
> Query: {fail1 query}
> Response: {fail1 response}
> Ground Truth: {fail1 gt}
> Failure Case 2:
> Query: {fail2 query}
> Response: {fail2 response}
> Ground Truth: {fail2 gt}
> Failure Case 3:
> Query: {fail3 query}
> Response: {fail3 response}
> Ground Truth: {fail3 gt}
> (3) Based on the comparison between the correct and failed demonstrations, explain what kind of reasoning behavior is essential for robust and correct student model performance. You should explicitly go through each failure case one by one using the ground-truth answer, identify the correct reasoning pattern, uncover any implicit assumptions present in the correct reasoning path that lead to the correct final answers, and analyze how to correct the incorrect reasoning chains.
> Then, new instruction prompts are designed to: (a) preserve the reasoning pattern exhibited in the correct demonstration; (b) incorporate the reasoning strategies and implicit assumptions necessary to reach the correct answers in the failure cases; and (c) guide the language model to self-identify and self-correct when similar failure patterns arise, steering it toward the correct reasoning path.
> Please perform the entire thinking process described above within the <prompt_think>...< /prompt_think> tag.
> Please output the modified instruction prompt clearly inside <generated_prompt>...< /generated_prompt> tags.
> Please do not include any demonstration example inside <generated_prompt>...< /generated_prompt>.
> You should be aware that every query has a valid answer. So the generated prompt must encourage the language model to conclude a valid answer.
> Do not include unrelated words such as < |im_start| > inside <generated_prompt>...< /generated_prompt>, just instructions for solving the problem.

### B.2 EXAMPLES OF SYNTHETIC REASONING PRIMITIVE

**Example primitives for the iGSM Dataset.** The generated primitive examples for solving this math dataset are shown in the following boxes with simplification for display. Example 1 aims to prevent errors where the model jumps to answers without properly resolving dependencies or dropping intermediate relationships. It addresses these issues by requiring the model to explicitly construct the dependency graph, clearly identify the target value to solve, and conduct a more detailed exami-

nation, including recomputing key steps even when no errors are detected. Example 2 focuses on preventing misinterpretation of the given conditions or overlooking parts of the problem description. For instance, it helps the model avoid skipping constraints or variables that may appear redundant. The most distinctive parts in Examples 1 and 2 are highlighted in red and blue, respectively. All generated primitives are combined with format instructions to ensure the teacher model generations comply with the required output format.

---

**iGSM Example prompt 1**:
Problem-Solving Instructions for Robust Reasoning
1. Define Variables and Equations:
- List every entity and attribute mentioned in the query.
- Assign a unique variable to each quantity.
- Translate all given statements into equations using these variables.
2. Prioritize Independent Variables:
- Solve variables with direct numerical assignments first (e.g., constants like 'X = 5').
- Substitute these values into the dependent equations immediately.
3. Build Dependency Graphs:
- For each unsolved variable, list all equations where it appears.
- Identify the simplest equation to resolve next (e.g., least dependencies).
4. Solve Step-by-Step:
- Proceed incrementally, substituting known values into dependent equations.
5. Validate Intermediate Results:
- Recompute critical steps to verify consistency.
- Flag contradictions for re-evaluation.
6. Target-Focused Resolution:
- Clearly state the goal variable.
- Back-substitute from the goal to ensure all required variables are resolved.
7. Final Answer:
- Conclude with a boxed numerical answer ('boxed{N}') supported by the validated reasoning chain.
Note: All variables are solvable.

---

**iGSM Example prompt 2**:
Instructions for Solving the Problem:
1. Define All Variables Explicitly:
- Assign variables to each entity and relationship mentioned, including composite terms (e.g., "Enclosure" = sum of sub-components).
- Label all variables clearly (e.g., ( X_Store ) for "X at Store").
2. Translate Statements into Equations:
- Convert every given statement into a mathematical equation, even if it seems redundant.
- Preserve all operations (sums, differences, multipliers) exactly as stated.
3. Substitute Known Values Early:
- Substitute fixed values (e.g., "equals 9") immediately to simplify equations.
- Do not assume unconstrained variables are zero unless explicitly stated.
4. Explore Indirect Relationships:
- Track how variables influence others indirectly (e.g., if ( A = B + C ) and ( B ) depends on ( D ), express ( A ) in terms of ( D )).
- If a variable's value is unresolved, check if it can be expressed in terms of other known variables.
5. Self-Correct for Consistency:
- If equations lead to contradictions (e.g., ( 0 = 1 )), revisit earlier steps to identify missed relationships or incorrect assumptions.
- Verify that all given statements are fully utilized in the solution.
6. Conclude Rigorously:
- Ensure every step logically follows from the previous ones.
- If ambiguity remains, exhaustively test plausible interpretations to understand the problem and conditions (e.g., "Enclosure" as a sum) to arrive at a valid answer.

---

## C  USAGE OF LLMS

We primarily used publicly available LLMs to assist with proofreading and grammar refinement of the paper draft. All technical content was verified by the authors. Our research also involves LLMs as a core component: we distill LLMs to curate datasets and also fine-tune LLMs. The authors take full responsibility for all research ideas, technical contributions, and conclusions.

