# OpenReview forum: "Tailored Primitive Initialization is the Secret Key to Reinforcement Learning"
_ICLR.cc/2026/Conference — ICLR 2026 Conference Withdrawn Submission_

### Official Review · Reviewer_6JyM · 2025-10-17

**Soundness:** 2
**Presentation:** 2
**Contribution:** 2
**Rating:** 4
**Confidence:** 4

**Summary:**

The paper advocates to discover diverse reasoning primatives so that the coverage of these tokens are existing in the output distribution of the downstream model. This way the warmstart data will incentivize the student model to have good coverage over reasoning primatives, which in turn have better exploration/performance after RL training. They first analyze the coverage problem, then introduce their finetuning pipeline and evaluate their approach against baseline methods. For their proposed Tailor pipeline, they rollout the student model and first let the teacher model analyze the traces and produce primatives that can better answer the question. Then prompt the teacher again to utilize these primatives during demonstration rollouts. Finally SFT on the collected dataset from these demonstrations.

**Strengths:**

- the method is more tailored to the student model, because the teacher model analyze from actual traces from the student models traces, rather than the traditional warm start where the teacher model just produce a static SFT dataset given some prompts
- the topic is relavant, the writing is mostly clear

**Weaknesses:**

- more costly than static dataset because need to analyze traces by the teacher model. Also relies on the teacher model to correctly analyze the reasoning traces to produce valid and improving reasoning primatives. Overall, the method is more reliant on utilizing and relying on the potentially expensive teacher model

**Questions:**

- have you tried using the student model itself as the one producing the primatives? and potentially condition on that and do RL/expert iteration on it?
- you apply filtered data collection on iGSM, but CoT and your way of prompting the teacher should have different accuracy right? what were the numbers?
- can you share some primatives that are commonly generated by the model?
- what is the effect of varying the teacher model? I would think your method is very reliant on the teacher's capability of proposing primatives and acting on them
- does the primatives vary across the 25 generations given the same prompt? for example is it 1 generate you first generate the primatives then generate the response, or you generate once the primatives, fix that and generate rest of the responses?
- you mentioned that the decoding for teacher is 0.5 in the experiment settings section, but in ablation temprature has better performance at > 0.5, e.g., 0.7/1.2. How come these two values were not used but 0.5 was chosen?

---

### Official Review · Reviewer_JaBj · 2025-10-25

**Soundness:** 3
**Presentation:** 3
**Contribution:** 2
**Rating:** 4
**Confidence:** 3

**Summary:**

This paper frames “reasoning-primitive coverage” as the driver of sampling inefficiency and initialization sensitivity in warm-start RL. It introduces Tailor: analyze failures, synthesize diverse primitives, curate multi-path high-quality demonstrations to widen the thinking-token distribution, then apply SFT and RL (DAPO/GRPO). On iGSM and KK across Llama3.2 and Qwen2.5 in both ID/OOD, Tailor yields faster convergence and higher accuracy; ablations attribute gains to the scale/diversity of primitives.

**Strengths:**

1. Grounded in a thinking-token (reasoning-primitive) coverage view, the paper optimizes the SFT stage with Tailor, an automated pipeline that analyzes failures and synthesizes repair-oriented primitives to curate diverse demonstrations—explicitly broadening the reasoning-state distribution before RL and thereby improving exploration during training.
2. The experiments are comprehensive: experiments on KK and iGSM across Llama3.2-1B/3B and Qwen2.5-0.5B/3B use DAPO with a rule-based reward and compare against Rule-based CoT, Standard-CoT, 4-STaR, and Batch-GKD (BGKD-λ); Tailor achieves the strongest results and faster learning on both in-distribution and OOD splits.
3. A diversity analysis of the curated SFT traces (via NV-Embed-v2 similarity) shows lower pairwise similarity under Tailor, indicating greater strategy-level diversity that better seeds RL exploration.
4. Ablations substantiate the design: increasing the number of primitives boosts post-RL accuracy; teacher decoding temperature benefits RL up to a point (over-high temperatures harm trace quality); and the gains persist under GRPO, underscoring algorithmic robustness.

**Weaknesses:**

1. The main results center on KK and iGSM, which are relatively simple suites, while validation on widely recognized harder benchmarks is missing, for example, MATH or AIME-24/25.
2. This paper primarily uses Llama/Qwen base models and selects DeepSeek-V3 as the teacher model; however, current RL practice often targets reasoning models such as DeepSeek-R1 and its distilled variants. Since these models already exhibit stronger self-reflection in chain-of-thought, it remains unclear whether the proposed method still offers superior gains compared with directly sampling chain-of-thought from a reasoning model.
3. The diversity of the curated SFT data is measured via embedding similarity. More behavior-level metrics might better capture dataset diversity and its impact on learning, such as SFT pass@K and the average thinking-token length.
4. Tailor introduces a multi-stage pipeline during SFT data construction, which likely incurs substantial token and compute overhead.

**Questions:**

1. Could you report performance on more difficult benchmarks, such as MATH or AIME-24/25, under the same settings used in the paper?
2. If the teacher is replaced with a stronger reasoning model (e.g., DeepSeek-R1 or a distilled variant), does Tailor still yield clear benefits over directly distilling from that reasoning model?
3. Figure 4 shows different starting points across methods. Are the SFT datasets identical in composition and training settings, with the only difference being the chain-of-thought? If so, could you provide quantitative indicators such as SFT pass@K and average token length to more clearly compare SFT dataset construction across methods?
4. Can you provide token costs for constructing the SFT datasets across different methods?

---

### Official Review · Reviewer_H94X · 2025-11-01

**Soundness:** 2
**Presentation:** 2
**Contribution:** 2
**Rating:** 2
**Confidence:** 4

**Summary:**

The paper argues that warm-starting RL-tuned LLMs with diverse, high-quality reasoning primitives improves sample efficiency and downstream performance. It introduces Tailor, a finetuning pipeline that (i) analyzes student traces, (ii) synthesizes repair-oriented primitives with a teacher LLM, and (iii) curates an SFT dataset conditioned on these primitives before RL. Experiments on iGSM (grade-school math) and K&K logical puzzles across Llama-3.2 (1B/3B) and Qwen-2.5 (0.5B/3B) show higher post-RL accuracy for Tailor versus baselines.

**Strengths:**

- The pipeline is simple and effective.

**Weaknesses:**

- The main idea of the paper is to use a teacher model to produce enhanced prompts that are concatenated with the original questions, and these augmented inputs are then used to construct SFT data for training the student model. However, this approach looks more like prompt engineering combined with data distillation rather than a fundamentally new learning paradigm. The so-called primitives are essentially reasoning strategies written as prompts, which the model is forced to imitate during SFT. This potentially harms the model’s ability to form its own reasoning process during the later RL stage and conflicts with the goal of enabling the model to autonomously generate CoT from the question alone. Since the DeepSeek-R1 results, the community increasingly believes that models should only be given the task itself while allowing CoT to emerge implicitly through learning, rather than embedding handcrafted strategies as part of the prompt. In this sense, the contribution of the paper feels limited and somewhat misaligned with current trends in reasoning research.

- The description of the TAILOR pipeline is not sufficiently clear, especially in the implementation details. For example, line 219 states that an LLM-based teacher is employed to correct erroneous trajectories, but in the pseudocode the second step is described as “Summarize Failures.” It is unclear whether correcting error trajectories and summarizing failures are the same operation or two sequential steps. The pseudocode also introduces k and m for failure summaries and primitive sets, yet the ranges or specific values for these variables are not provided. It is not stated whether every trajectory undergoes failure summarization and primitive generation, or whether these steps are performed only after processing a certain number of samples. There is also no explanation of how many trajectories are summarized each time, how many samples are used to generate one primitive, or whether there are any ablation studies on these design choices.

- Moreover, in the training stage it is unclear whether primitives are treated as part of the response and thus receive gradient updates, or if they are only included in the prompt without being learned by the model. If primitives are not part of the model’s output space and no gradient flows through them, then the student model never learns to generate primitives and will rely on externally provided primitives at inference time. In that case, every test instance would still require sampling a primitive to construct the prompt; otherwise there would be a distribution mismatch between training and inference. The paper does not clarify how this issue is handled, which makes it difficult to evaluate the validity and robustness of the method.



- The study evaluates only KK and iGSM. So it remains unclear how Tailor translates to broader, real-world tasks . Moreover, experiments use small models (0.5B–3B) without scaling trends, limiting conclusions about behavior at larger scales. Add more reasoning tasks (e.g., MATH MMLU, GPQA) alongside KK and iGSM to test external validity, and report a basic scaling curve including 7B/14B models to assess capacity interactions with Tailor.

- Provide qualitative case studies: the same question solved under several primitives with commentary on when each helps.

**Questions:**

See Weaknesses

---

### Official Review · Reviewer_HSeh · 2025-11-07

**Soundness:** 3
**Presentation:** 3
**Contribution:** 3
**Rating:** 6
**Confidence:** 4

**Summary:**

This paper addresses two key challenges in applying reinforcement learning (RL) to large language models (LLMs): low sample efficiency and inconsistent performance across models. The authors attribute these issues to the models’ strong sensitivity to initialization data—particularly, the lack of diverse and high-quality reasoning primitives during the warm-start stage. To tackle this, the paper proposes Tailor, a fine-tuning pipeline that analyzes the failure trajectories of student models using a teacher model, summarizes and automatically synthesizes diverse and learnable reasoning patterns, and constructs a more comprehensive SFT dataset to enhance subsequent RL exploration and final performance. Experiments on mathematical and logical reasoning benchmarks show that Tailor achieves higher sample efficiency compared to rule-based CoT, 4-STaR, and re-distillation baselines.

**Strengths:**

1.The paper focuses on the problem of data construction during the warm-start stage of reinforcement learning and proposes the Tailor pipeline, which centers on reasoning primitives to automatically select high-quality and diverse training data. The idea is novel and insightful.
2.The experimental design is relatively complete. Experiments on the iGSM and K&K benchmarks with small Llama and Qwen models provide sufficient evidence for the method’s effectiveness and reproducibility.
3.The paper is clearly written and easy to follow.
4.The work offers a feasible data-centric perspective for improving RL sample efficiency and cross-model stability, showing meaningful research and application value.

**Weaknesses:**

1.Terminology and formalization are insufficient. The notion of thinking token coverage is ambiguous (it can be read as “diversity of reasoning patterns” or as a “token proportion”). If there is an accepted definition, please cite it explicitly in Related Work; if not, provide a computable formal definition and measurement protocol, and add derivations/claims in the preliminaries that directly support the core thesis.
2.Motivation and positioning are not strong enough. The Introduction and Related Work do not sufficiently establish the necessity and gap for “warm-start data construction,” which weakens later horizontal comparisons. Suggested revisions: add concrete failure cases and cost analyses (sample efficiency, training instability) in the Introduction; in Related Work, systematically review data-centric RL initialization and primitive-based literature, clarify differences, and expand comparisons across alternative data construction strategies and datasets.
3.Insufficient evidence for “discovering new primitives.” The abstract claims automatic discovery of novel reasoning primitives, but the body reads more like identification/combination of existing patterns. Please provide criteria for novelty (e.g., similarity thresholds to existing corpora/templates, blinded human agreement), a catalog of new primitives, and ablation showing performance drop when each purportedly new primitive is removed.
4.Limited coverage. Experiments are restricted to iGSM and K&K and mainly small models, which limits claims of generality. Please explain the selection principles (synthetic/verifiable, low contamination risk, cost control) and add broader validation; if compute is constrained, report equal-token-budget comparisons showing unit-cost gains over baselines.
5.Lack of an end-to-end example. Examples are scattered across sections, making practical reproduction difficult. Add an appendix with a minimal end-to-end reproducible case.
6.Reliability and failure modes are not reported. The paper does not quantify pipeline success rates, failure scenarios, or mitigation strategies, nor does it describe trustworthiness/filters for generated data. Please report pass rates, failure type distributions, and the safety/quality filtering stack (plus any human audit statistics).
7.Compute cost and scalability are underreported. There is no quantitative accounting of token costs for teacher reasoning, primitive synthesis, and data generation, nor a discussion of bottlenecks and optimizations at larger scales. Include a stage-wise cost table and outline scaling paths and limits for larger models/datasets.

**Questions:**

See the "Weaknesses" section.

---

### Note · Authors · 2025-12-13

I have read and agree with the venue's withdrawal policy on behalf of myself and my co-authors.